# Regaining policy attention for a health insurance capitation payment reform in Ghana: A prospective policy analysis

Gilbert Abotisem Abiiro [1,2]*

1 Department of Health Services, Policy, Planning, Management and Economics, School of Public Health, University for Development Studies, Tamale, Ghana, 2 Department of Population and Reproductive Health, School of Public Health, University for Development Studies, Tamale, Ghana

* gabiiro@uds.edu.gh

**Data Availability Statement:** All relevant data are included in this published article in the form of direct quotations from the respondents. The qualitative dataset (tape recordings and transcripts) contains potentially sensitive information and

## Abstract

Capitation as a provider payment mechanism gained policy attention by the Ghana National Health Insurance Scheme (NHIS) in 2012 and was piloted in the Ashanti Region, Ghana. Recent studies revealed that the policy was suspended in 2017 due to inappropriate policy framing, actor contestations, unclear policy design characteristics, and an unfavorable political context. However, the NHIS still has interest in capitation as a provider payment option. Using the modified political process model, a prospective policy analysis was conducted to explore how to: i) appropriately reframe policy debates; ii) create political opportunities; and iii) mobilize resources to reattract policy attention to capitation in Ghana. Cross-sectional qualitative data were gathered in December, 2019 from semi-structured interviews with a purposive sample of 18 stakeholders and complemented with four community-level focus group discussions with 41 policy beneficiaries in the pilot region. All data were tape-recorded and transcribed. The analysis was thematic, using the NVivo 12 software. The results revealed that an appropriate reframing of the policy requires policy renaming, refinement of certain policy design characteristics (emergency care, capitation rates, choice and assignment of providers) and refocusing policy communication and advocacy on the health benefits of capitation instead of its cost containment intent. To create political opportunities for policy re-implementation, a politically sensitive approach with broader stakeholder consultations should be adopted. Policy advocacy and communication should be evidenced-based and led by politically neutral agents. An equitable capitation policy implementation requires resourcing health facilities, especially the lower-level facilities, with improved infrastructure, consumables, improved information management systems and well-trained personnel to enhance their service delivery capacities. The study concludes that there exists stakeholder interest in the capitation policy in Ghana and calls for an effective reframing, creation of political opportunities, and mobilization of needed resources to regain policy attention.

personal identifiers that permission was not obtained from the participants and the ethics review committee for public sharing. Queries about public access to qualitative data may be directed to the Ghana Health Service Ethics Review Committee (ethics.research@ghsmail.org or +233302681109).

**Funding:** This work was supported by a family foundation grant administered through a fellowship award to the author (GAA) by the Center for Policy Impact in Global Health, Duke Global Health Institute, United States of America. The funder had no role in study design, data collection and analysis, decision to publish, or preparation of the manuscript.

**Competing interests:** The author has declared that no competing interests exist.

## Background

Reforming healthcare provider payment mechanisms to incentivize healthcare providers is essential for ensuring quality healthcare within low- and middle-income countries (LMICs) [1–3]. Capitation, one of the prospective provider payment mechanisms, involves advanced fixed payments per client made by a purchaser, to a healthcare provider to cover the healthcare costs of clients registered under the provider for an agreed period of time [2, 4]. Capitation can create positive incentives for healthcare providers to be accountable and efficient in the use of healthcare resources, control escalating healthcare costs, encourage disease prevention and health promotion, facilitate facility-level planning and budgeting, and enhance existing health care gatekeeper systems [1, 4–6]. It is, therefore, often portrayed by health care purchasers, such as governments and health insurance authorities, as a good alternative to enhance strategic purchasing that will ensure value for money [1, 7]. On the contrary, capitation produces negative incentives for healthcare providers to provide fewer health services, encourages prematured discharge of patients and high referral of cases to higher level facilities, leading to overburdening of higher-level facilities and poor access to quality health care [3]. These varied incentives of capitation for stakeholders have implications for the political feasibility and long-term sustainability of such reforms. Such reforms, by nature, become political, creating serious implications for the interest of political actors and the popularity of political regimes [8, 9].

It is, therefore, often difficult to gain political priority or policy attention for controversial policy reform options such as capitation [8, 10–12]. The existing literature demonstrates that even if such policy options manage to get onto the policy agenda, they risk survival in the short term [8, 13–15]. There is evidence of a number of policy options falling off the policy agenda at the initial stages of policy initiation and formulation, during a pilot phase of policy implementation, during a scaling up phase or few years after commencement of full implementation of the policy [8, 13, 16]. The fall of such policies have often been attributed to inappropriate framing of policy ideas and debates, poor policy design characteristics, actor contestations, unfavourable political environment, resource constraints, poor stakeholder involvement, poor communication on policy, unfavourable contextual peculiarities and other implementation challenges [8, 11, 17–20]. These challenges can severely hinder the ability of such reforms to regain policy attention within the contexts in which they were adopted/implemented. Generally, there is limited evidence on the possibility of policies that have lost political priority to regain attention for re-implementation. Additionally, the actions required to bring back a policy that has fallen off the policy agenda onto the implementation table have not been adequately explored.

In Ghana, capitation gained policy attention by the National Health Insurance Scheme (NHIS) in 2012 and was piloted in the Ashanti Region of Ghana until its suspension in 2017 [8]. Escalating health care costs to the NHIS due to the inherent incentives created by the existing fee-for- service (FFS) and the Ghana Diagnostic Related Group (G-DRGs) payment systems facilitated the creation of a policy window for capitation [21]. The NHIS legislation also acknowledged capitation as one of the provider payment options available to the scheme [22]. This legal backing of capitation and the cost sustainability issues confronting the NHIS at the time resulted in an initially wide acceptance of the concept of capitation by the policy community of the NHIS [8, 11]. It was, therefore, adopted and then placed under a pilot implementation in the Ashanti region to generate evidence for a nation-wide scale up [11]. Abiiro et al. [8] argued that the choice of the Ashanti region for the pilot in 2012 was technically justified due to its central location, heterogenous culture and varied health system characteristics, but that decision was associated with a huge political risk since the region was the stronghold of the then opposition political party (New Patriotic Party ([NPP]), which became the ruling

government since 2017. A detailed description of the design characteristics [23] and pilot implementation of the NHIS capitation policy [8] as well as assessments of the initial effects of the policy have been documented elsewhere [24–28].

In brief, the pilot capitation policy allowed NHIS clients in the Ashanti region to be assigned to one preferred primary care provider (PPP) selected from a list of three providers proposed by the client [23]. Clients were encouraged to choose providers within their vicinities (districts) of residence and could only change their PPPs after six months of registration [23]. Clients, however, complained that in some instances, they were not informed of the exact facilities that were assigned to them as their PPPs [8]. Each provider was paid a monthly fixed amount (capitation rate) per client to deliver the capitation benefit package which mainly constituted primary health care (PHC) (medications and services) at outpatient departments. Inpatient care at higher level referral facilities, specialised and emergency care were still purchased using the G-DRGs and FFS mechanisms [23]. While providers argued that the capitation rate was low and they did not understand how it was determined, the NHIS clients reported challenges getting access to emergency care which was supposed to be excluded from the benefit package of the capitation policy [8].

Using the Shiffman and Smith [29] policy analysis framework, Abiiro et al. [8] analysed the pilot implementation of the Ghanaian capitation policy to understand the factors that pushed it off the policy agenda. Abiiro et al. [8] revealed that in principle, there was widespread support for the concept of capitation, however, poor framing and communication of the policy, inappropriate and unclear policy characteristics, stakeholder opposition, and an unfavourable political environment interacted to cause the suspension of the capitation policy. A summary of the relevant issues that contributed to the suspension of the policy [8] are presented in Table 1. With the suspension of the capitation policy, the NHIS maintains the G-DRGs for purchasing out-of-patient and inpatient services and FFS for purchasing medications nationwide [21, 22]. Since the suspension of the policy, the NHIS still recognises capitation as a viable policy option for consideration. Such consideration, however, requires evidence to inform policy redesign and implementation. Although studies in Ghana have provided insights into the reasons behind the suspension of the policy [8, 11, 30], further evidence is required to understand how political priority/policy attention can be regained on the capitation policy in Ghana. Adopting the Geneau et al. [31] modified political process model, this study sought to answer the questions; what is required to appropriately; i.) reframe the capitation policy debates, ii.) identify and create political opportunities and iii.) mobilise resources to reattract policy attention to capitation in Ghana. The evidence generated from this study contributes to assisting policy makers, including the NHIS, to understand the political feasibility of a re-implementation of the capitation policy in Ghana.

## Methods

### The policy analytical framework

The political process model was developed to assess the determinants of political priority on a policy issue, specifically relating to the activities of social movements [32]. In the field of health policy, it was adapted and modified by Geneau et al. [31] to examine the conditions that contributed to low global priority on chronic diseases by development partners and to recommend strategies to raise political priority for chronic diseases in LMICs. According to the modified political process model, raising policy attention on an issue requires; i) reframing the debates on the issue, ii) identification and creation of political opportunities, and iii) resource mobilization. Reframing the debates requires changes in the conceptualization, labelling and issue presentation to reshape the myths, beliefs, attitudes and emotions associated with the policy

Table 1. Summary of the issues that led to the suspension the NHIS capitation policy in Ghana.

| Policy issue | Key findings |
| --- | --- |
| **Framing** | • Framing capitation as a cost containment strategy to the neglect of portraying the medical benefits of capitation in policy communication<br>• Framing capitation as a fraud prevention strategy, thereby, labelling health care providers within the pilot region as fraudsters<br>• Framing capitation as a restrictive policy on access to health care<br>• Framing the choice of Ashanti region for the pilot as unethical since pilots are normally carried out in relatively smaller locations<br>• Framing capitation as a discriminatory political strategy against an opposition stronghold |
| **Actor power dynamics** | • Poor stakeholder understanding of the pilot policy<br>• Varied and conflicting stakeholders' interests in the policy<br>• Opposition to the choice of Ashanti region for the pilot<br>• Political actors' dominance and influence over policy implementation<br>• Emergence of an anti-policy alliance of actors<br>• Mass protest against policy through organized press conferences, public demonstrations and petitions<br>• Poor mobilization and involvement of the civil society |
| **Policy characteristics and implementation issues** | • Inadequate public education on the policy<br>• Challenges with certain policy design characteristics (i.e assignments of clients to providers, emergency care and membership portability)<br>• Exclusion of medicines from the capitation basket<br>• Policy incompatibility with existing health system characteristics<br> • Poor health care gate keeper system<br> • Limited prescription responsibilities of lower-level health facilities<br> • Uneven geographical distribution of health facilities and personnel<br>• Delayed in pilot completion and nationwide scale-up<br>• Unintended negative consequences of the policy<br> • Increasing health care cost<br> • low patronage of PHC facilities<br> • Increasing out-of-pocket payments,<br> • Increasing mortalities due to restrictive health care access |
| **Political context** | • Choice of an opposition stronghold as region for pilot gave rise to politics<br>• Low trust in then ruling government (NDC) in the pilot region<br>• Politicians (party communicators) involvement in policy communication<br>• Opposition members of parliament from the region withdrew support for the policy<br>• Then opposition party campaigned against policy during the 2016 elections' campaigns<br>• Government (National Democratic Congress [NDC]) that introduced capitation lost election and votes in the capitated region<br>• Change in government in 2017<br>• Anti- capitation policy campaigners appointed into the new NPP government |

(Source: Abiiro et al. [8])

issue and its intent [8, 12, 33]. Appropriate reframing can generate positive stakeholder attitudes towards the policy [8]. Identification and creation of political opportunities require deliberate attempts to take advantage of favorable global and national events/meetings that open windows of opportunities for the policy [29, 34]. It also entails deliberately putting in

place systems, structures, processes and mechanisms that will build consensus around a policy issue and attract widespread support for the policy [34]. The last component of the model requires that appropriate human, material, financial and technical resources be mobilized and made available to support efforts to raise political attention on a policy issue [12, 31]. The modified political process model has been used to analyse health policy issues at the global level [12, 31]. The application of this model to analyse a national level policy is, therefore, innovative. Besides, the application of the Geneau et al. [31] model fits very well within the objective of the current study which aims at recommending what will be required to regain political priority for capitation in Ghana.

## Study setting, design and sampling

This cross-sectional study, adopting a prospective perspective, was implemented in two districts in the Ashanti region where the capitation policy was piloted. A detailed description of the study context, design, sampling and methods of data collection have been documented in Abiiro et al. [8]. The Kumasi Metropolis and the Ahafo Ano South district constituted the study sites. These districts were purposively selected to reflect the urban (Kumasi Metropolis) and rural (Ahafo Ano South district) disparities of the Ashanti regional population.

The respondents comprised key stakeholders of the NHIS capitation policy located at the regional, district or community levels in the Ashanti region. The study purposively identified the key institutions/organisations that had interests in the capitation policy and/ played various roles in the implementation of the policy [8]. These comprised health care providers and administrators, political parties, pressure groups, labour unions, civil society organisations, and policy beneficiaries, who were all recruited in December, 2019. Each stakeholder institution was contacted on phone or through a visit to the organisation and the persons that were directly involved in the implementation of the policy were identified and purposively selected for inclusion in the study. The participants comprised the leadership/representatives of the following organisations: Ashanti regional health directorate (n = 2), incumbent political party (NPP) (n = 1), opposition political parties (National Democratic Congress [NDC]) (n = 1), academics and researchers (n = 1), Ghana Registered Nurses and Midwives Association (n = 1), Society of Private Medical and Dental Practitioners (SPMDP) (n = 1), coalition of non-governmental organizations (NGOs) in Health (n = 1), Ashanti Development Union (ADU [regional pressure group]) (n = 1), district health directorates (n = 2), public health facilities (n = 3), private-for profit health facilities (n = 2), and Christian Health Association of Ghana (CHAG) facilities (n = 2). Besides these 18 key stakeholders, community leaders and gatekeepers assisted the researcher to recruit 41 community level policy beneficiaries, who were active members of the NHIS within the period of the capitation pilot (2012–2017) for inclusion in the study. Staff of the NHIS (a key stakeholder of the policy) at the regional and district levels refused to participate in the study because of the political sensitivity of the policy issue and the fact that they did not receive instructions from the National Health Insurance Authority (NHIA) to respond to the study.

**Ethics approval and consent to participate.**   Ethical clearance was obtained from the Ghana Health Service Ethics Review Committee (GHS-ERC009/10/19). Permission to conduct the study was obtained from the Ashanti Regional Health Directorate and all stakeholder organisations. Written informed consent was obtained from all respondents.

*Data collection and analysis*. The data were collected through interviews with the 18 key stakeholders and four community-level focus group discussions (FGDs) with policy beneficiaries in December, 2019. Interview and FGD guides were developed for a bigger qualitative study of which this paper is a component. Some of the findings from this bigger qualitative

study have already been published [8]. The interview and FGD guides for the overall study have been attached as S1 & S2 Files, respectively. For the purpose of this paper, the relevant parts of the FGD and interview guides required the respondents to comment on the future of the suspended capitation policy. The respondents were specifically asked their opinions about the re-implementation possibilities of the policy with probes to elicit their perceptions about the prospects and challenges for policy re-implementation. The last part of the data collection instruments requested respondents to make specific recommendations/suggestions that the capitation policy makers (NHIA) may consider to facilitate a re-implementation of the policy.

In each district, two gender-specific FGDs were conducted (one for males and one for females). The FGDs were organised in a purposively sampled community in each district [8]. The number of participants per FGD was between 9 and 12. The interviews were conducted using the English Language but the FGDs were conducted in Twi (the local language). Each interview lasted about 45 minutes and each FGD lasted about one hour. The FGD and interview guides were pre-tested. Member checking, through respondents' validation, was employed to enhance the credibility of the data [35]. Interviews and FGDs were conducted at serene places. Two research assistants supported the author to conduct the interviews and the FGDs in December 2019. The research assistants (postgraduate students) were very fluent in both English and the local language (Twi). To ensure consistency across data collection, the research assistants were trained on the study protocol, instruments, procedures and ethics of data collection. All responses were audio-recorded and later transcribed. The data from the FGDs were translated from Twi into the English Language by the two research assistants. Peer checking, where the research assistants reviewed the translations of each other, was deployed to enhance the accuracy of the translations.

The data were analysed using the NVivo 12 software. The thematic analysis approach was employed with the main and subthemes identified through deductive and inductive coding of the transcripts, respectively. The three tenets of the Geneau et al. [31] policy analysis framework provided the conceptual basis for deductive coding whilst the specific subthemes emerging from the broad themes were identified through progressive inductive coding. The author coded all the transcripts using the NVivo software. To enhance confirmability in the coding and derivation of themes, two working colleagues of the author, who are very conversant with the NVivo software, reviewed and provided feedback on the codes generated. Table 2 presents the main and subthemes identified from the analysis of the data. Thick descriptions of the themes with the aid of direct quotations from the transcripts has been deployed to provide a context of the findings and a voice to the respondents. Overall, the study was implemented according to the Principles of the Declaration of Helsinki.

## Results

### Reframing the capitation policy

**Renaming the policy.**   The respondents argued that since the policy was suspended bearing the name "capitation", it implies that stakeholders have rejected the word "capitation" as a policy label. However, once there is persistence and/ renewed interest in the policy issue, especially from the NHIA, healthcare providers, NGOs and health administrators, the politicians may consider reprioritizing it for implementation. The respondents, however, argued that the policy elites are currently reluctant in bringing back the same policy that was suspended because that may be interpreted as a betrayal of the public. Acknowledging the relevance of capitation within the Ghanaian NHIS context, the general consensus among stakeholders was that a different policy label or name that conveys similar intent as capitation should be given to the policy before it is brought back for implementation.

**Table 2. Recommendations of the stakeholders on regaining policy attention for capitation.**

| Issue (main theme) | Recommendations from stakeholders (subthemes) |
|---|---|
| **Reframing** | • Renaming the policy<br>• Refining the policy content<br> • Capitation of medicines<br> • Realistic capitation rates<br> • Clarity in assignment of clients to PPPs<br> • Multiple choices of facilities for PPPs<br> • Effective coverage of emergency care<br>• Refocus policy communication on medical benefits of capitation but not cost containment |
| **Creating political opportunities** | • Adopt a politically sensitive approach to re-implementation/piloting<br> • Once off nationwide implementation<br> • Repilot in a politically balanced region<br> • Resume with multiple pilots across the country<br> • Adhere to proper principles of piloting<br>• Demonstrate evidence of a thorough review of the pilot implementation<br>• Broader stakeholder consultation and involvement<br>• Evidenced-based policy communication led by politically neutral agents<br> • Academics/ researchers<br> • Civil society organisations<br> • Community-based organisations and groups<br> • Healthcare providers<br> • Local leaders |
| **Resource mobilization** | • Resourcing rural health facilities to ensure equitable capitation implementation with:<br> • improved infrastructure and consumables<br> • well-trained personnel<br> • accommodation for health workers<br>• Reviewing the prescription responsibilities of lower-level health facilities<br>• Strengthening of healthcare referral system with:<br> • readily available ambulances and<br> • proper information management systems |

"*The name capitation itself has been slaughtered so, if they can come with a different thing apart from capitation but do it in a way that people will not think that, they are bringing something at the back door but just changing the name*"

(Administrator, Regional Health Directorate)

"*The Society of Private Medical and Dental Practitioners have been making suggestions of re-introducing it. The politicians will also not want to eat back their words and bring it back, so they are looking, should we change the name*?"

(Administrator, Regional Health Directorate)

**Refining the content of the policy.** The respondents further cautioned against merely changing the name of the policy without refining its content to demonstrate to stakeholders that the new policy is somehow better than the previous one.

*"I can tell you that most people will want it back but they will want it someway refined"*

(Administrator Regional Health Directorate)

Some of the respondents, especially academics and health care administrators, recommended the inclusion of medicines into the capitation benefit package at all levels of the health care system if the policy will have a positive effect on controlling healthcare cost. They, however, indicated that including medicines into the benefit package should be associated with an increase in the capitation rate. They believe that the capitation rates that were paid during the pilot phase were not realistic because the amounts were deemed insufficient to cover the cost of services that were provided. Health care providers and administrators argued that an attractive capitation rate will be required to motivate all providers to embrace a reintroduction of the policy.

*"They should add medicines into the capitation benefit package to control cost and increase the capitation rate small to cover the inclusion of the medicines"*

(Administrator, CHAG, health Center, Ahafo Ano South)

*"Also, they should make the capitation amount in such a way that it will help entice every provider to join. For now, I think they are paying me GH¢ (Ghana Cedi) 12–15 per visit (G-DRGs rate), so if they peg the capitation rate per person at even GHC10 or GH¢12, it will attract providers to join"*

(Administrator, CHAG, Health Center, Ahafo Anor South)

Furthermore, respondents argued that there is the need to ensure clarity in how PPPs are assigned to clients and once the assignment is done, there should be an effective way of communicating to clients about the facilities assigned to them. The participants added that since the Ghana Health Service has naturally ranked facilities according to levels of care—hospitals, health centers and CHPS compounds, NHIS clients should be given the opportunity to choose and be assigned to three different facilities (one hospital, one health center and one CHPS). Also, there should be an effective network of linkages of the clients' information across these three facilities. Once a referral path is established, clients are able to initiate care at the first level of their PPP, and can get referrals to the next level without constraints.

*"Every smaller facility should have been linked to a bigger facility as a referral centre. If that thing was added to the policy, like they (PHC) will provide the first aide there and if they feel it is too much then they refer you to a particular place. That should be a better way to go"*

(Coalition of NGO in Health).

*"I think already they have ranked all the health facilities because we have the CHPS system, we have the health centres, we have the district hospitals, the teaching hospitals. So at least they should add two facilities to the main core facility that every client selects as his primary point of health service delivery. So that, if you have the primary facility, the next facility would be higher so that when I am sick, the moment I go to this facility and they are to refer me, it means I am going to the next facility. I think when they do that it will help"*

(Administrator, District Health Directorate, Ahafo Ano South)

As already reported (Table 1), although the capitation policy excluded access to emergency care from its benefit package, some clients reported being denied services during emergency. Some providers were aware that the costs of emergency services were supposed to be reimbursed through other payment mechanisms but argued that the policy was not very clear on what constituted an emergency. Providers made their value judgments as regard what situation qualified as an emergency which often resulted in the denial of some emergency care seekers access to services. The respondents, therefore, recommended that the policy should ensure effective coverage of emergency care through clarity in the definition of emergencies and the modalities of seeking emergency care under the policy. Providers and clients should be educated on the procedures for accessing emergency care under the policy.

'*Also, they have to ensure that the facilities are also playing their role because the capitation policy did exclude emergencies [. . .] but I think some of the facilities when the clients went there and it was emergency, they were not helping the clients, they were just driving them away when they check their machine and found out that, they did not choose that facility for the capitation*"

(Administrator, Public Hospital, Kumasi)

**Refocusing policy communication on the medical benefits of capitation but not cost containment.** All participants agreed that policy communicators failed to appropriately educate the public on the health benefits of capitation. It was, therefore, recommended that to attract wide support, especially from the policy beneficiaries, communication on the rationale of a capitation policy should be refocused on the health care benefits that stakeholders stand to gain from the policy but not the cost-containment and fraud prevention motives of the policy.

"*if you ask me what can be done, I will tell you what I always hammer on, proper communication. Let the patients themselves understand that it is in their own interest that you are doing this for them. Let them understand that instead of hopping around, it is even medically proper, it is very good, to have one person taking care of you and then keeping your records. In that case, your history and other stuff are there, so remain there until when the person or the facility feels that looking at your condition, they have to refer you, they then refer you to a specific place which they think in their system, they can control your issues for you.*

(ADU)

"*Communication on the policy should be improved. For people who really understand capitation, all the benefits, those advantages of capitation, were not made clear in the policy communication [. . ..] Instead of telling the (health) benefits of capitation, they were saying capitation was going to address faults [. . .]. Though, the objectives of capitation are splendid, the communication on the policy was the problem*

(Administrator, Regional Health Directorate)

"*If they have the opportunity to re-implement it, they should let the education go down well and explain it to people the benefits one stands to get when they select one hospital for you, this is how it will help you and this is what you will benefit from it*"

(FGD, Male, Kumasi)

### Identification and creation of political opportunities for policy re-implementation

**Adopt a politically sensitive approach to policy re-implementation.** The respondents argued that a politically sensitive approach to policy re-implementation is required to build consensus for a successful policy re-implementation. Firstly, the stakeholders argued that to avoid the usual perception of regional political discrimination in policy implementation and to portray the policy as a national policy, the best approach to policy re-implementation is to resume the implementation with a once off nation-wide implementation without the need for repiloting, especially in the Ashanti region.

> "*If it is rolled out full time in the whole country, nobody will complain because that is the policy now and everybody will be getting one policy that is being rolled out. But to start it again in the Ashanti region, it will be very difficult*
>
> (Administrator, CHAG, Health Center, Ahafo Ano South)
>
> *If it will be only in the Ashanti region again, it will be very difficult because people may still think that it is discrimination [. . .]But if it is rolled out in the whole country, I think it will be much easier to get support*
>
> (Administrator, Private hospital, Ahafo Ano South)

On the other hand, the respondents argued that if the NHIS will like to keep to the gradual approach of scaling up from pilot implementation, then the policy implementation should resume with a repiloting in a politically balanced region such as the Bono region or Ahafa region etc. Alternatively, there could be multiple pilots across the geopolitical space of Ghana in order to generate hybrid political and contextual evidence to inform a nationwide scale-up. The respondents cautioned that despite the approach to repiloting that will be adopted, the policy implementers must adhere to proper principles of piloting by ensuring that the pilot phase is restricted to a relatively smaller area and completed within a relatively shorter, well-defined and well-communicated timeframe.

> "*First and foremost, they shouldn't start it in the Ashanti region again or in the Volta region. These are the strongholds of the two political parties [. . .] because politics is like a cancer that was eating in it. So, it should be done in a quiet region so that we can learn a lot from there so that gradually we can expand. We can add one region at a time*"
>
> (Academic)
>
> "*Anything that comes and faces challenges and by virtue of that it is suspended will not have an easy come back and so if there should be a comeback, I think that they should adhere to proper piloting principles and do it on a very small scale. You do it on the small scale to attain success then based on that success, you begin to scale up. That is the only way that we can convince the bigger picture that this thing is feasible, it is sustainable and beneficial*"
>
> (Medical Director, CHAG, Hospital, Ahafo Ano South)

**Demonstrate evidence of a thorough review of the pilot implementation.** The respondents argued that stakeholders would welcome back the policy, if the NHIA demonstrates evidence of an extensive review and evaluation of the pilot phase in the Ashanti region. Based on this evaluation, the NHIA should clearly demonstrate to stakeholders the effects (positives and

negatives) of the policy, the implementation challenges faced during the pilot, the lessons learnt from the pilot and the steps that are being proposed to address the pilot-related challenges.

> "*I think that if they want to re-implement it, it will be good provided they will let us sit down and review the way it was done during the pilot*"
>
> (Administrator, Regional Health Directorate)

> "*Like, I keep repeating, there was a program, the program was implemented in a section of the population. Before you can conclude on its positives, you need to do an evaluation first on the implementation over the period*"
>
> (Politician, NDC)

> *I support the suspension of the policy if only they will actually take the advantage of the suspension to do an evaluation and check the bottlenecks and carve a way forward.*"
>
> (Coalition of NGOs in Health)

**Broader stakeholder consultation and involvement.** It was recommended that the NHIA should consult and involve all important stakeholders in the redesign and re-implementation of the policy. They argued that when stakeholders are deeply involved in the process, they will gain knowledge about the policy, co-own the policy and contribute to a successful implementation of it.

> "*I always think that capitation is good. It is the way they did it which wasn't good. So, if they will let us sit down and some of us who understand the concept will really get involved, bring our ideas for them to modify it, it will be better because it reduces waste in the system*"
>
> (*Administrator, Regional Health Directorate*)

**Evidenced-based policy communication led by politically neutral agents.** The respondents argued that empirical evidence from the evaluation of the pilot phase should be the basis for policy communication on the rationale and benefits of capitation. The respondents argued that education on the policy should be led by politically neutral agents such as staff of the NHIA, academic/ researchers, civil society organisations, community-based organisations, pressure groups, healthcare providers, traditional and religious leaders, among others. Public education on the policy should be adequately extended to the community level, relying on appropriate community-based information dissemination channels such as community radios, durbars and religious fora created by mosques and churches.

> "*They should still hold on to the suspension and educate the citizens for them to understand it well. They should educate us on how capitation is going to benefit us, help us to understand it well and let us know when they did the pilot what were the results they got before they take any decision again on it*"
>
> (FGD, Male, Kumasi)

> "*I think the politicians should also back-off from it and let those who have the ideas, the policy makers develop the policy and educate the public about the policy*"
>
> (Administrator, CHAG, health center, Ahafo Ano South)

### Resource mobilization

**Resourcing rural health facilities to ensure equitable capitation implementation.** The respondents argued that those who live in rural areas and/ receive care from lower-level healthcare (PHC) facilities including CHPS compounds and health centers were disadvantaged in the implementation of the capitation policy. The respondents, especially policy beneficiaries at the community level, strongly recommended that for rural residents and lower-level health facilities to accept capitation in the future, adequate financial, material and human resources should be channeled by the Ministry of Health to improve upon the service delivery capacities of PHC facilities through the provision of improved infrastructure and consumables, and well-trained and well-accommodated health professionals.

"*With capitation, I think before you implement it you have to put in place adequate mechanisms by providing health facilities, doctors' bungalows and extending electricity to all places to ensure that all facilities can deliver quality health care [. . .] When government is able to do all these, then they can re-implement the capitation policy but if they cannot then we don't need the capitation*"

(FGD, Ahafo Ano South)

**Strengthening of healthcare referral system with readily available ambulances and proper information management.** The respondents argued that strengthening the existing healthcare gate keeper system with ambulances to facilitate referral to higher level facilities in times of emergencies and complications can significantly build trust in the capitation system when it is re-implemented. The policy beneficiaries argued that they will be willing to choose the facilities at their community levels as their PPPs if they were guaranteed of readily available transportation from those facilities to the higher-level facilities in case of a serious illness requiring referral. The stakeholders also argued that for effective capitation, there is the need for an effective health care information management system that will ensure effective linkages of the information systems of the various health facilities to facilitate an effective referral process when the need arises.

"*If this capitation will work, then ambulances should be stationed at the village level health facilities. In case of emergency and I have a CHPS compound as my PPP and I go there first and my sickness gets serious but they don't even have ambulances, so what time will I go to a lorry station and get a car to go to the facility they will refer me to?*"

(FGD, male, Ahafo Ano South)

"*The country has to ensure effective information management before re-introducing capitation. For example, when I am diagnosed at Komfo Anokye teaching hospital of a heart problem and I travel to Dormaa and I attend a hospital there, the doctors there should be able to access my information at Komfo Anokye from a common platform*"

(FGD, Male Kumasi)

**Reviewing prescription responsibilities of lower healthcare facilities.** The respondents, especially health care providers and community level policy beneficiaries argued that lower-level health facilities were not attractive to NHIS clients as PPPs because the Ghana health system regulations do not allow those facilities to prescribe a number of medications including

antibiotics. The respondents, therefore, argued that for capitation to work at this level, there is the need for the Ghana Health Service to review this policy to allow more responsibilities to health workers at the lower levels to dispense more medications than currently allowed for them.

> "*The only thing is that the Ghana Health Service should do some modifications to its regulations especially in the health centres to allow them to prescribe a lot of drugs that they are currently not supposed to write. This is because when some of the conditions happen in the night and the PPP cannot initiate treatment, the person may die on his way to the referral facility. So at least they should start something before referral so that when they get to the hospital, it might have been controlled*"

(Medical Assistant, Public District hospital, Ahafo Ano South)

## Discussion

Using the Geneau et al. [31] modified political process model, the study has established the possibility of the suspended Ghanaian NHIS capitation policy to regain policy attention. To realise this potential, the study has highlighted from the perspective of key stakeholders what is required to reframe the policy debates, create political opportunities and mobilise needed resources to reattract policy attention.

The finding that an appropriate reframing of the capitation policy will require renaming the policy, refining the policy content and refocusing policy communication on the health benefits of capitation is very important. This is because poor policy framing contributed to public misinterpretations of the policy intent during the pilot implementation [8]. The name of a policy can create a long lasting cognitive label in stakeholders [33] that re-introducing a policy with a previously "rejected policy name or label" can trigger stakeholders' resistance of the policy. The call from stakeholders to replace the name capitation with an alternative label that conveys comparable policy intent is therefore justified. Although no specific renaming nomenclature was recommended by stakeholders, capitation falls within the range of provider payment mechanisms labelled valued-based payment models [36]. Issahaku et al. [7] assessed the feasibility of value-based payment within the NHIS of Ghana and recommended it for policy consideration. As revealed from the study, renaming of the policy should be accompanied with content refinement of the policy to demonstrate to stakeholders that the fresh policy is an improved version of the previously piloted policy. Policy refinement should affect adjustments in the capitation rate, and education on its determination and how it differs from the rates currently paid under the G-DRGs and FFS systems. This will reduce providers' misinterpretations of the capitation rate as synonymous to that of the existing G-DRGS and FFS rates [8]. The possibility of forming networks of providers to enable clients to be capitated to three providers at different levels of the health care system is also worth consideration in policy design. Standard conceptualisation, procedural consensus and public education on medical emergency under capitation are required. Policy framing is an interpretive issue that is influenced by people's socio-economic, ideological and professional backgrounds and experiences [37]. It is, therefore, essential that the framing of policy rationale should resonate the perspective of the audience but not just technical design intentions. Policy advocacy and communication on the capitation policy should, therefore, convey targeted messages rather than generic technically framed messages. For instance, capitation can be framed as an effective strategy to ensure predictable revenues for effective planning and budgeting targeting healthcare providers while

issues relating to continuity of care, closeness with healthcare providers, accurate health record keeping and proper diagnosis should target clients. The health implications of capitation should, therefore, be central in policy advocacy to regain policy attention. As argued by Bertone et al. [37], to create a common and shared understanding of policies, policy framing should be flexible where the actual (technical) frames reflecting design intents of the policy could be metaphorically presented in ways that reflect common and shared contextual understandings, values, beliefs and expectations of the policy.

The main finding relating to the identification and creation of political opportunities for policy re-implementation centred on building consensus among stakeholders to generate widespread support for the policy. As a requirement for building consensus, a politically neutral approach to policy implementation has been recommended, with two main implementation options. The stakeholders' opinions on these two options reflect an inherent debate about the relative significance of a once-off nation-wide policy implementation strategy viz-a-viz a gradual approach to policy implementation through scaling up implementation from piloting [38]. The arguments for a once-off nationwide implementation approach are premised on its ability to reduce the perception of geopolitical discrimination and political favouritism in policy implementation [8]. On the other hand, proponents of the gradual approach through piloting based their arguments on practical realism in the face of constraints on resources and administrative capacity, and the need to test policy effectiveness before full scale implementation [39, 40]. Although a once-off approach to policy re-implementation is most likely to have the greater effect on generating political consensus for the policy, the option of repiloting in a politically less sensitive region or relying on multiple pilots across the geopolitical setting of Ghana is also practically appealing. Based on the experiences of the pilot phase in the Ashanti region, repiloting of the policy must adhere to the essential principles of policy piloting. According to Ettelt et al. [40], a policy pilot" *involves policy implementation in a limited number of places, typically over a predefined period of time, coupled with a more or less formal approach to policy learning, usually, but not exclusively, through external evaluation".* A good pilot should therefore have a clear purpose, be restricted to a smaller geographic area, completed within a relatively shorter and clearly defined period of time and must be accompanied by an evaluation that will generate feedback to establish policy effectiveness [39]. It is argued that the suspension of the pilot has created an opportunity for policy evaluation to generate evidence on policy effectiveness and implementation feasibility as the basis of canvassing fresh political support for policy re-implementation. Interesting, a number of studies by academics/researchers have evaluated or examined various aspects of the capitation policy [8, 11, 17, 25, 26, 28, 30]. What is required is for the NHIA to pull this empirical scientific evidence on the policy pilot together in the form of a systematic review as the basis for advocating for fresh political support for the policy. There is an urgent need for the NHIA to complement the individual academic research works by commissioning series of studies to assess the prospects and challenges of the capitation policy. The evidence generated from these studies should be disseminated at national stakeholder dialogue fora to engender wider policy debates on the capitation policy. This scientific evidence should be the bases for policy communication, education and advocacy for political attention on the policy. This advocacy for the policy should be led by politically neutral stakeholders such as the NHIA, academics, civil society organisations, pressure groups and health care providers. Deliberate efforts should be made to prevent partisan communicators' involvement in education on the capitation policy.

A well organised, structured, supplied and regulated health system with a well-developed PHC structure provides an essential context for the implementation of a capitation policy [41]. It was against this background that stakeholders recommended the strengthening and resourcing of the Ghana health system and regulatory policy reforms to strengthen the service

delivery capacities of PHC facilities as a pre-requisite to regain attention for the capitation policy. It is worth noting that these issues fall within the functional scope of the Ministry of Health and that of the Ghana Health Service and may be beyond the scope of the NHIA. The successful implementation of a capitation policy, therefore, requires collaboration between the NHIA (purchaser) on the one hand and the Ghana Health Service and the Ministry of Health on the other. The current weak healthcare gate keeper system that facilitates bypassing of PHC facilities must be addressed to make capitation effective [42]. This requires equipping PHC facilities with adequate infrastructure, personnel and ambulances to facilitate effective first line treatment and referral of patients when needed. There is also the need to improve upon the information management system within the health system to facilitate sharing of patient records across health care facilities. To make PHC facilities attractive as PPPs for clients, it is suggested that regulatory reforms by the Ghana Health Service should enable more basic medications to be administered at this lowest level of the health system.

## Strengths and limitations of the study

The study adopted an innovative application of a policy analysis framework within the policy context of an LMIC, and therefore contributes to filling gaps in the limited application of theory in health policy analysis, especially within LMICs. Besides, this is the first application of the Geneau et al [31] modified political process model to sub-national level policy analysis. The study is timely as it complements the works of Abiiro et al. [8] and others to provide a comprehensive perspective on much-needed evidence to support policy decision on a health care provider payment reform (capitation) in Ghana. The global health policy community can draw key lessons from this Ghanaian experience. It is, however, acknowledged that, as a qualitative prospective analysis of a specific policy issue within the specific Ghanaian context, caution is required in drawing implications to inform similar policy reforms within similar contexts. The failure of the staff of the Ghanaian NHIS to respond to the study implies that a key stakeholder perspective was not adequately captured. Finally, given the political sensitivity of the policy issue, future changes in the political context of the study setting can have significant influences over the long-term applicability of the study findings.

## Conclusion

The study has demonstrated that regaining policy attention for capitation in Ghana entails reframing the policy debates by renaming the policy, refining the content of the policy and refocusing policy communication on the health benefits of capitation. Policy education should be refined to reflect clarity in policy provisions for emergency care, capitation rates, and the procedures for assignment of PPPs. The creation of political opportunities for policy re-implementation require consensus building through the adoption of a politically sensitive approach to policy advocacy, communication and implementation. Evaluation evidence from the pilot phase of the policy is required to engender effective policy communication and advocacy led by politically neutral agents with broader stakeholder consultation and involvement. Resource mobilisation for policy re-implementation should entail strengthening and resourcing the Ghana health system and regulatory policy reforms to enhance the service delivery capacities of PHC facilities for an equitable capitation policy implementation. The study acknowledges the existence of stakeholder interest in the capitation policy and calls for an effective reframing, creation of political opportunities, and mobilization of needed resources to reattract attention to the policy in Ghana.

## Supporting information

**S1 File. Interview guide.**
(PDF)

**S2 File. FGD guide.**
(PDF)

## Acknowledgments

The author acknowledges the support of Professor. Roger A. Atinga and Professor Kennedy A. Alatinga for reviewing the data analysis and drafts of the manuscript. The support of Faisal Abdallah Kaamah and Mohammed Salifu during the data collection is also acknowledged.

## Author Contributions

**Conceptualization:** Gilbert Abotisem Abiiro.

**Data curation:** Gilbert Abotisem Abiiro.

**Formal analysis:** Gilbert Abotisem Abiiro.

**Funding acquisition:** Gilbert Abotisem Abiiro.

**Investigation:** Gilbert Abotisem Abiiro.

**Methodology:** Gilbert Abotisem Abiiro.

**Project administration:** Gilbert Abotisem Abiiro.

**Resources:** Gilbert Abotisem Abiiro.

**Software:** Gilbert Abotisem Abiiro.

**Validation:** Gilbert Abotisem Abiiro.

**Writing – original draft:** Gilbert Abotisem Abiiro.

**Writing – review & editing:** Gilbert Abotisem Abiiro.

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
