## [Decision Letter · Decision Letter 0]

8 Mar 2024

PGPH-D-23-02465

Regaining policy attention for a health insurance capitation payment reform in Ghana: A prospective policy analysis

Dear Dr. Abiiro,

Thank you for submitting your manuscript to PLOS Global Public Health. After careful consideration, we feel that it has merit but does not fully meet PLOS Global Public Health’s publication criteria as it currently stands. Therefore, we invite you to submit a revised version of the manuscript that addresses the points raised during the review process.

We look forward to receiving your revised manuscript.

Kind regards,

Miquel Vall-llosera Camps

Staff Editor

Journal Requirements:

Reviewers' comments:

Reviewer's Responses to Questions

**Comments to the Author**

1. Does this manuscript meet PLOS Global Public Health’s publication criteria? Is the manuscript technically sound, and do the data support the conclusions? The manuscript must describe methodologically and ethically rigorous research with conclusions that are appropriately drawn based on the data presented.

Reviewer #1: Yes

Reviewer #2: Yes

2. Has the statistical analysis been performed appropriately and rigorously?

Reviewer #1: N/A

Reviewer #2: N/A

3. Have the authors made all data underlying the findings in their manuscript fully available (please refer to the Data Availability Statement at the start of the manuscript PDF file)?

Reviewer #1: Yes

Reviewer #2: Yes

4. Is the manuscript presented in an intelligible fashion and written in standard English?

Reviewer #1: Yes

Reviewer #2: Yes

5. Review Comments to the Author

Reviewer #1: Thank you for affording me the opportunity of reviewing this manuscript. I believe that that authors have addressed a very pertinent issue, that of funding healthcare to facilitate access to healthcare for all. This manuscript is well written. The authors have argued very well for the study. The aims of the study are very clear. The findings resonate well with aims and are supported by clear excepts from participants. There is a clear link between the analysis framework and the findings. The discussion is sounds and the conclusions made are solid. Even though the manuscript is overall well written the following does need to be addressed before it can be accepted for publication:

1. The study design does need to be reflected in the abstracts and so does the number of FG participants. Also check the length of the abstract. It is quite lengthy and wordy in some areas.

2. Participants demographics need to be included as part of the methodology section of the study.

3. On page 8, the section on "study setting, design and sampling" the authors mention in the first sentence that "Respondents comprised of key stakeholders of the NHI" however in the last 3 sentences of the paragraph they say that "Staff of the NHI refused to participate.....". This sounds contradictory. So which one is it?

4. Only credibility through member checking has been accounted for in trustworthiness, what about the other strategies?

5.Explain how data collected in Twi was handled during the analysis. Was it translated? How did you ensure that the translations are correct ?

6.How did you ensure consistency of data collection across interviews and focus groups especially because data was collected by the authors and 2 assistants, so different people?

7.Who did the coding? How did you ensure reliability during this process? How were codes and ultimately themes arrived at?

Reviewer #2: Authors undertake a prospective policy analysis of a planned reform process in Ghana. It is an important piece of evidence and would be a great learning opportunity for countries planning to re-implement a policy or re-focus policy attention to a reform that has lost political focus post pilot implementation. It is very well written, and I have no recommendations in terms of any major changes.

I recommend on editorial change in the background on the sentence below:

. “.. the region was the stronghold of the then opposition political party (New Patriotic Party ([NPP]), which is now the ruling government.” The “now” in this sentence needs to be removed or be qualified by stating the year in which NPP is in power. I think the point you are trying to make is that the party that was in opposition then (and probably opposed the pilot) is now in power and is meant to superintend over the proposed re-implementation. The authors could be more explicit about this and then you would not have to mention the year.

6. PLOS authors have the option to publish the peer review history of their article (what does this mean?). If published, this will include your full peer review and any attached files.

**Do you want your identity to be public for this peer review?** For information about this choice, including consent withdrawal, please see our Privacy Policy.

Reviewer #1: No

Reviewer #2: **Yes: **Brendan Kwesiga

---

## [Decision Letter · Decision Letter 1]

2 May 2024

Regaining policy attention for a health insurance capitation payment reform in Ghana: A prospective policy analysis

PGPH-D-23-02465R1

Dear Dr. Abiiro,

We are pleased to inform you that your manuscript 'Regaining policy attention for a health insurance capitation payment reform in Ghana: A prospective policy analysis' has been provisionally accepted for publication in PLOS Global Public Health.

Best regards,

Julia Robinson

Executive Editor

Reviewer Comments (if any, and for reference):

Reviewer's Responses to Questions

**Comments to the Author**

1. If the authors have adequately addressed your comments raised in a previous round of review and you feel that this manuscript is now acceptable for publication, you may indicate that here to bypass the “Comments to the Author” section, enter your conflict of interest statement in the “Confidential to Editor” section, and submit your "Accept" recommendation.

Reviewer #2: All comments have been addressed

2. Does this manuscript meet PLOS Global Public Health’s publication criteria? Is the manuscript technically sound, and do the data support the conclusions? The manuscript must describe methodologically and ethically rigorous research with conclusions that are appropriately drawn based on the data presented.

Reviewer #2: Yes

3. Has the statistical analysis been performed appropriately and rigorously?

Reviewer #2: N/A

4. Have the authors made all data underlying the findings in their manuscript fully available (please refer to the Data Availability Statement at the start of the manuscript PDF file)?

Reviewer #2: Yes

5. Is the manuscript presented in an intelligible fashion and written in standard English?

Reviewer #2: Yes

6. Review Comments to the Author

Reviewer #2: None

7. PLOS authors have the option to publish the peer review history of their article (what does this mean?). If published, this will include your full peer review and any attached files.

**Do you want your identity to be public for this peer review?** For information about this choice, including consent withdrawal, please see our Privacy Policy.

Reviewer #2: No
